# Phenolics Profiling by HPLC-DAD-ESI/MS^n^ of the Scientific Unknown *Polygonum hydropiperoides* Michx. and Its Antioxidant and Anti-Methicillin-Resistant *Staphylococcus aureus* Activities

**DOI:** 10.3390/plants12081606

**Published:** 2023-04-10

**Authors:** Ygor Ferreira Garcia da Costa, Eulogio José Llorent-Martínez, Laura Silva Fernandes, Pedro Henrique Santos de Freitas, Elita Scio, Orlando Vieira de Sousa, Paula Cristina Castilho, Maria Silvana Alves

**Affiliations:** 1Department of Pharmaceutical Sciences, Faculty of Pharmacy, Federal University of Juiz de Fora, Juiz de Fora 36036-900, Brazil; ygor.fgc@gmail.com (Y.F.G.d.C.); fernandes_ls@outlook.com (L.S.F.); ovsousa@gmail.com (O.V.d.S.); 2Department of Physical and Analytical Chemistry, Faculty of Experimental Sciences, University of Jaén, Campus Las Lagunillas s/n, E-23071 Jaén, Spain; ellorent@ujaen.es; 3Department of Biochemistry, Biological Sciences Institute, Federal University of Juiz de Fora, Juiz de Fora 36036-900, Brazil; pedrofreitasufjf@gmail.com (P.H.S.d.F.); elita.scio@ufjf.br (E.S.); 4Madeira Chemical Center, Faculty of Exact and Engineering, University of Madeira, Campus da Penteada, 9020-105 Funchal, Portugal

**Keywords:** *Polygonum hydropiperoides*, gallic acid derivatives, flavonoids, antioxidants, antibacterial agents, methicillin-resistant *Staphylococcus aureus*

## Abstract

*Polygonum hydropiperoides* Michx. is an Asian native plant species that is also widely distributed in the Americas. Despite its traditional uses, *P. hydropiperoides* is scarcely scientifically exploited. This study aimed to chemically characterize and investigate the antioxidant and antibacterial activities of hexane (HE-Ph), ethyl acetate (EAE-Ph), and ethanolic (EE-Ph) extracts from aerial parts of *P. hydropiperoides*. The chemical characterization was performed through HPLC-DAD-ESI/MS^n^. The antioxidant activity was assessed by the phosphomolybdenum reducing power, nitric oxide inhibition, and the β-carotene bleaching assays. The antibacterial activity was determined by the minimal inhibitory concentration (MIC) and the minimal bactericidal concentration followed by the classification of the antibacterial effect. Chemical characterization revealed the expressive presence of phenolic acids and flavonoids in EAE-Ph. An increased antioxidant capacity was revealed in EAE-Ph. Regarding antibacterial activity, EAE-Ph showed weak to moderate property against 13 strains tested with MIC values ranging from 625 to 5000 µg/mL, with bactericidal or bacteriostatic effects. Glucogallin and gallic acid stand out as the most relevant bioactive compounds. These results suggest that *P. hydropiperoides* is a natural source of active substances, supporting this species’ traditional use.

## 1. Introduction

The World Health Organization (WHO) declared antimicrobial resistance (AMR) as one of the major, serious, and complex public health problems of the 21st century, and gave recommendations for the research, discovery, and development of new antibiotics, mainly for the treatment of multidrug-resistant (MDR) bacteria, specially to the bacterial species that are most commonly isolated from infections around the world and that express resistance genes. Among them, as a high-priority pathogen, methicillin-resistant, vancomycin-intermediate, and vancomycin-resistant *Staphylococcus aureus* needs to be noteworthy [1,2]. These strains are related to skin and soft tissue infections, particularly the complicated forms of these processes [3].

Furthermore, the study of the antioxidant activity of compounds represents an important field because the deregulation between the oxidative and antioxidant molecules, with an increased oxidative one, impacts cell homeostasis. The formation of reactive species, such as reactive oxygen species (ROS) and nitrogen species (RNS), and the lipid peroxidation mediated by the formation of free radicals play a significant role in human diseases, being associated with many pathological conditions such as cardiovascular and neurodegenerative disorders. Additionally, the occurrence of oxidative stress-mediated by infections, such as bacteremia evolving to sepsis, can promote fatal consequences. The inhibition of this process can induce fewer toxic effects in these cases and could contribute to the survival of the patients [4,5,6,7].

So, to attend the WHO appeal and contribute to this scenario, researchers around the world bring back attention to natural products (NPs) as potential sources of bioactive compounds. In fact, NPs and their derivatives are promising sources of substances, originating many drugs used in clinical practice. Newman and Cragg [8] reported that of the 1881 drugs approved from 1981 to 2019, 787 (41.8%) are NPs or their derivatives.

In this context, Brazil stands out, having the major biodiversity worldwide, with about 22% (45,000 species) of world flora [9]. The Asian native plant species *Polygonum hydropiperoides* Michx. (Polygonaceae Juss), popularly known in Brazil as “erva-de-bicho”, is widely distributed. Traditionally, it is used as an antidysenteric, anti-hemorrhagic, anti-hemorrhoid, astringent, diuretic, and vermicide against varicose veins, erysipelas, rheumatism, and in the treatment of wounds [10,11].

Even though there are about 1000 *Polygonum* species, only about 30 are described as used ethnobotanically and only 21 were phytochemically investigated. Thus, a wide field of research remains open, and the identification and isolation of new active molecules (and their mechanisms of action) from other species would be of scientific merit [12]. In the Narasimhulu, Reddy, and Mohamed review, the occurrence of anthraquinones, coumarins, stilbenoids, flavonoids, phenylpropanoids, lignans, sesquiterpenoids, triterpenoids, and tannins as recognized chemical constituents in *Polygonum* species was documented [12]. It is important to point out that *P. hydropiperoides* was not mentioned in this literature survey. Flavonoids were found the most common components, being used as chemotaxonomic markers of this genus, playing an important role in the systematics of the Polygonaceae family species [13].

Only three publications on the chemical composition of *P. hydropiperoides* were found, reporting the presence of coumarins, steroids, or triterpenes, flavonoid heterosides, polyphenols, saponins, tannins, ref. [14] the organic acids gallic, malic and tannic, the anthocyanin pelargonidin, the flavonoids luteolin, quercetin and rutin, phytosterine, polygodial [10], melilotic, and *m*-hydroxybenzoic acids [14,15].

The present study aims to chemically characterize and investigate the in vitro antioxidant and antibacterial activities of hexane (HE-Ph), ethyl acetate (EAE-Ph), and ethanolic (EE-Ph) extracts from aerial parts of *P. hydropiperoides*.

## 2. Results and Discussion

### 2.1. Chemical Characterization

Phytochemicals were tentatively identified and characterized based on high-performance liquid chromatography with a diode array detector and an ion trap mass spectrometer with an electrospray interface (HPLC-DAD-ESI/MS^n^) spectral information, analytical standards, and literature data. The negative ion mode was used in all cases due to its high sensitivity for phenolics determination, although positive ion was also used, observing the same phenolic profile. Base peak chromatograms of EAE-Ph and EE-Ph extracts are shown in Figure 1 and Figure 2. HE-Ph was not chemically characterized, because this extraction aimed only to remove fatty acids and resins from the vegetable material, and it did not reveal antibacterial activity. Compounds were numbered according to their order of elution, keeping the same numeration in both extracts.

As displayed in Figure 1 and Figure 2, it was possible to detect 39 peaks in the EAE-Ph and EE-Ph chromatographic profiles. Among them, 32 (82%) could be tentatively identified as shown in Appendix A.

#### 2.1.1. Phenolic Acids

Compound **3** was tentatively identified as gallic acid based on its deprotonated molecular ion at *m*/*z* 169 and base peak at *m*/*z* 125. Compound **2**, with more 162 Da, corresponded to galloyl glucose. Compound **29**, with 169/125 fragmentation, was tentatively identified as a gallic acid derivative.

Compound **9** displayed the neutral loss of 162 Da to yield dihydroxybenzoic acid at *m*/*z* 153, [16] being tentatively identified as a dihydroxybenzoic acid-hexoside.

#### 2.1.2. Flavonoids

Compounds **4** and **6**, with [M − H]^−^ at *m*/*z* 577, exhibited the base peak at *m*/*z* 425 and fragment ions at *m*/*z* 451, 407, 289, and 287, in agreement with procyanidin dimers of the type (Epi)catechin–(epi)catechin [17]. Compound **12**, with an additional 152 Da (galloyl moiety) was tentatively identified as a procyanidin dimer monogallate [two units of (epi)catechin plus a galloyl moiety].

Compound **8** was characterized as catechin by comparison with an analytical standard. Compound **17**, [M − H]^−^ at *m*/*z* 441, suffered the neutral loss of 152 Da (galloyl moiety) to yield (epi)catechin at *m*/*z* 289, so it was tentatively identified as (epi)catechin-*O*-gallate.

Compound **10** suffered the neutral loss of 162 Da (hexoside) to yield dihydrokaempferol at *m*/*z* 287 (main fragment ion at *m*/*z* 259) [18].

Compounds **14** and **15** suffered neutral losses of 132 and 146 Da, respectively, which corresponded to pentoside and deoxyhexoside moieties. The aglycone was present at *m*/*z* 317, with fragments at *m*/*z* 271 and 179 (myricetin), so they were tentatively identified as myricetin glycosides.

Compound **32** presented deprotonated molecular ion at *m*/*z* 331 and base peak at *m*/*z* 316. This fragmentation had been previously reported for mearnsetin [19].

Compound **33** was identified as quercetin ([M − H]^−^ at *m*/*z* 301 and fragment ions at *m*/*z* 179 and 151) by comparison with an analytical standard. Seven quercetin derivatives were present in the analyzed extracts. Compounds **16**, **19**, **20**, **21**, and **22** were tentatively identified by the following neutral losses: 162 Da (hexoside), 132 Da (pentoside), 176 Da (glucuronide), 146 Da (deoxyhexoside), 204 Da (acetylhexoside; 42 + 162 Da), and 174 Da (acetylpentoside; 42 + 132 Da).

Compound **39** was tentatively identified as isorhamnetin due to the 315→300 fragmentation, and fragment ions at *m*/*z* 271, 179, and 151. Compounds **18**, **23**, **27**, and **31** were isorhamnetin glycosides (identification based on the neutral losses previously discussed). Compound **26** also presented the 315/300 fragmentation. However, the additional fragment ions were not consistent with isorhamnetin, so its complete identification was not performed.

#### 2.1.3. Other Compounds

Compound **1** was tentatively identified as a disaccharide (two hexoses). It suffered the neutral loss of a hexoside moiety (162 Da) and the fragmentation pattern was characteristic of hexoses [20].

Compound **5** was tentatively identified as a phloroglucinol derivative. It exhibited neutral losses of a methyl group (−15 Da), methylphloroglucinol unit (−140 Da), galloyl moiety (−152 Da), and the gallic acid residue (at *m*/*z* 169) [21].

Compounds **35** and **38** were tentatively identified as oxo-dihydroxy-octadecenoic and trihydroxy-octadecenoic acids, considering data from the scientific literature.

These results are in agreement with Nikolaeva, Lavrent’eva, and Nikolaeva findings [22]. These authors studied the phenolic composition of the aerial parts of *Polygonum amphibium*, *Polygonum angustifolium*, *Polygonum aviculare,* and *Polygonum divaricatum* by HPLC, and reported the presence of caffeic, chlorogenic, gallic, *p*-cumaric, and protocatechuic acids, the glycosylated flavonoids avicularin, kaempferol glycoside, hyperoside, isoquercetin, and rutin and the aglycones quercetin and kaempferol. Furthermore, ferulic acid was detected in *P. amphibium* and *P. angustifolium*, luteolin and luteolin-7-glycoside in *P. aviculare* and *P. divaricatum*, and cosmosiin and apigenin in *P. amphibium*, *P. aviculare,* and *P. divaricatum*, which demonstrated the diversity of this chemical class of substances in this genus [22].

In addition, Quesada-Romero et al. [23] reported the presence of quercetin, isorhamnetin, and kaempferol and their derivatives in *Persicaria maculosa*, a plant species from the Polygonaceae family, also corroborating our findings.

Quantification of individual phenolic compounds was performed using UV chromatograms. Only the 18 most abundant ones were quantified, due to the low signal or overlapping observed in some peaks. The results are given in Table 1.

Table 1 shows the expressive presence of phenolic compounds reaching values of 95 and 56 mg/g of the dry extract of EAE-Ph and EE-Ph, respectively. Phenolic acids such as galloyl glucose or β-glucogallin (43 and 48 mg/g of dry extract in EAE-Ph and EE-Ph, in this order) and gallic acid (21 and 4.2 mg/g of dry extract in EAE-Ph and EE-Ph, respectively). Compound **2** (galloyl glucose or β-glucogallin) was responsible for approximately 85% and 45% of the total individual phenolic content (TIPC) present in EE-Ph and EAE-Ph, in this order. Compound **3** (gallic acid) contributed with 22% of the total phenolics in EAE-Ph and 7% in EE-Ph.

Phenolic compounds or polyphenols constitute a broad group of substances originating from the secondary metabolism of plants (it is estimated that more than 8000 structures are known) and can be found in different natural sources such as fruits, vegetables, nuts, seeds, stems, flowers, honey, and beverages such as coffee, tea, and red wine being responsible for pigmentation and astringency [24,25,26]. They also act as protective agents against the harmful action of UV light, in addition to protecting plant species against parasites and insects [27].

These substances can be present as simple molecules (benzoic and cinnamic acids) to highly polymerized compounds (lignin, melanin, and tannin), with flavonoids representing the most common and widely distributed subgroup [25].

As shown in Table 1, phenolic acids and flavonoids were identified in this study, and the concentrations of total phenolic acids (64.26 and 52.20 mg/g of dry extract in EAE-Ph and EE-Ph, respectively) were visibly higher than those of flavonoids (30.96 and 4.22 mg/g of dry extract in EAE-Ph and EE-Ph, in that order). In their study, Quesada-Romero et al., [23] when evaluating the composition of total phenolics and flavonoids in extracts from aerial parts of *P. maculosa* (Polygonaceae), obtained concentrations of total phenolics ranging from 9.6 to 14.3 (methanolic extracts) and 13.7 to 16.6 (ethanolic extracts) mg of gallic acid equivalents/100 g of extract; and a lower concentration of total flavonoids ranging from 6.6 to 8.4 (ethanolic extracts) and 8.2 to 8.6 (methanolic extracts) mg of catechin equivalents/100 g of extract, thus corroborating our findings.

Phenolic compounds have a broad range of bioactivities, from antioxidant and anticancer properties to the ability against bacterial strains [27,28]. Regarding to pathogenic bacteria, in recent decades, with the advancement of analytical tools, studies could be carried out with commercialized extracts and/or pure or synthetic phenolic compounds. In 2016, Lima et al. [25] evaluated the in vitro antibacterial activity of gallic, caffeic, and pyrogallol acids against clinical strains of *Escherichia coli* 06 (EC06; isolated from urine), *Pseudomonas aeruginosa* 15 (PA15; catheter tip), and *Staphylococcus aureus* 10 (SA10; rectal swab). According to these authors, the three phenolic compounds tested did not show clinically relevant antibacterial activity, with MIC values ≥ 1024 µg/mL.

The main phenolic compound detected in the tested extracts (43 and 48 mg/g in EAE-Ph and EE-Ph dry extract, in that order) was galloyl glucose or β-glucogallin (Table 1). Pawłowska et al. [29] analyzed the antibacterial and anti-inflammatory activities of the aqueous extract from roots of *Bistorta officinalis* Delalbre [*Polygonum bistorta* (L.) or *Persicaria bistorta* (L.) Samp], a perennial plant also belonging to the Polygonaceae family, and its major phytochemicals, derived from galloyl glucose and flavan-3-ols. Galloyl-glucose derivatives and catechin derivatives were effective against *S. aureus* [(ATCC 6538), (ATCC 25293) and (ATCC 43300)], *Staphylococcus epidermidis* (ATCC 14990) and clinical strains methicillin-resistant *S. aureus* (MRSA) 13318 and methicillin-resistant *S. epidermidis* (MRSE) 13199, with MIC values ranging from 1 to 125 µg/mL (6-*O*-galloylglucose and 1,6-*O*,*O*-digalloylglucose) and 0.5 to 500 µg/mL [procyanidin B3, (+)-catechin,(-)-epigallocatechin gallate-(4,8)-(+)-catechin, (-)-epicatechin gallate-(4,8)-(+)-catechin, (-)-epicatechin-(4,8)-(-)-epicatechin gallate, (-)-epigallocatechin-(4,8)-(-)-epicatechin gallate, (-)-epicatechin gallate, (-)-epicatechin gallate-(4,8)-(-)-epicatechin gallate and (-)-epicatechin-(4,8)-(-)-epicatechin gallate-(4,8)-(-)epicatechin], respectively. These authors concluded that galloyl glucose and flavan-3-ols contribute to the general effect of the infusion, justifying the traditional use of *B. officinalis* [29].

Gallic acid (3,4,5 trihydroxybenzoic acid) is the second most abundant compound in EAE-Ph and EE-Ph extracts (21 and 4.2 mg/g of dry extract in EAE-Ph and EE-Ph, respectively).This bioactive phytochemical is often present in the human diet and is, therefore, considered “safe” and natural in the context of the food production system and used as a preservative in foods and beverages due to its power to scavenge free radicals (antioxidant capacity); in fact, gallic acid is used in the manufacture of gallic acid methyl ester, gallic acid lauryl ester, and propyl gallate, which are widely used antioxidant food additives [30,31]. In addition to its well-established antioxidant capacity, gallic acid has other pharmacological properties like other phenolic acids, such as antimicrobial, anticancer, anti-inflammatory, and antiulcerogenic [25,31]. It is interesting to notice that several flavonoids identified in *P. hydropiperoides* are esterified with gallate moieties (Appendix A. Indeed, this high amount of free gallic acid can be endogenous of the plant or it can be an artifact resulting from the de-esterification of galloyl-substituted catechins and other compounds during the extraction process, or both.

Flavonoids were detected at concentrations of 30.96 and 4.22 mg/g of dry extract in EAE-Ph and EE-Ph, in this order (Table 1). Probably, this expressive difference between EAE-Ph and EE-Ph contributed to the antibacterial effectiveness of EAE-Ph in detriment of EE-Ph. Based on the well-known antibacterial activity of this phytochemical class, many modes of action were reported, especially, the action (damage) in the bacterial membrane.

Catechins (present in concentrations of 8.9 mg/g of dry extract in EAE-Ph) were previously described as acting in the bacterial membrane by the generation of ROS, while quercetin (present in concentrations of 15.13 mg/g of dry extract in EAE-Ph) acts decreasing the proton-motile force in *S. aureus* strains, increasing the membrane permeability. In both cases, the rupture and death of the bacterial cells were promoted [32].

### 2.2. In Vitro Antioxidant Activity

The in vitro antioxidant capacity of EAE-Ph, EE-Ph and the standards [gallic acid, quercetin, rutin, Trolox, and butylated hydroxytoluene (BHT)] were assessed by the phosphomolybdenum reducing power, nitric oxide (NO) inhibition, and β-carotene bleaching assays, as can be seen in Table 2.

The total antioxidant capacity was evaluated by the phosphomolybdenum reducing power, demonstrating that EAE-Ph and EE-Ph were able to reduce Mo (VI) to Mo (V). In comparison, EAE-Ph and EE-Ph (214.57 ± 23.99 and 230.82 ± 4.60 µg/mg of ascorbic acid equivalents (AAE), respectively) were statistically equal and more effective than the gallic acid standard (160.78 ± 17.04 µg/mg AAE). However, these extracts represent about 50% of the quercetin (424.70 ± 10.99 µg/mg AAE) ability to reduce Mo (VI) to Mo (V). Berwal et al. [33], when analyzing the total antioxidant activity of methanolic extracts of *Calligonum polygonoides* (Polygonaceae) with the phosphomolybdenum method, revealed an antioxidant capacity under this assay, corroborating our findings.

The inhibition of NO formation revealed that EAE-Ph possesses the most relevant activity, being more effective than the standards gallic acid, Trolox (vitamin E analogue) and the EE-Ph extract, which showed statistically equal values. In addition, corroborating our findings, Choi et al. [34] demonstrated a positive inhibition of the NO production by the methanolic extract from aerial parts of *Persicaria senticosa* (Polygonaceae) with values of IC_50_ equal to 71.8 ± 3.7 µg/mL.

Antioxidant activity was also assessed by the β-carotene bleaching assay. It showed that EAE-Ph (74.44 ± 5.17%) was the most effective sample tested, being better than the standards quercetin (43.33 ± 5.47%), rutin (27.78 ± 7.43%), and BHT (61.11 ± 5.33%) in inhibiting the lipid peroxidation. EE-Ph was less effective than EAE-Ph but was statistically equal to the quercetin standard. Similar results were found by Sahreen, Khan, and Khan [35] who demonstrated the capacity of lipid peroxidation inhibition of methanolic extract and fractions from roots of *Rumex hastatus* (Polygonaceae). These authors showed that some fractions possess activity comparable with the standard catechin.

Together, these results revealed that EAE-Ph and EE-Ph possess an interesting antioxidant capacity being able to act in different mechanisms and inhibit oxidative stress. Probably this activity is related to the expressive presence of phenolic acids and flavonoids (Table 1), especially in EAE-Ph, the most effective one, that exhibits higher concentrations of these metabolites.

### 2.3. In Vitro Antibacterial Activity

The in vitro antibacterial activity of HE-Ph, EAE-Ph, EE-Ph, ampicillin (AMP), and chloramphenicol (CHL) was established by determining MIC and MBC values and the classification of the antibacterial effect as bactericidal or bacteriostatic, which are expressed in Table 3. The appropriate controls were performed to ensure that solvents used in the extraction and in the preparation of the solutions had no influence in the bacterial growth, demonstrating that at the tested concentrations it was not able to promote the bacterial inhibition.

MIC values for HE-Ph, EAE-Ph, and EE-Ph ranged from 625 to 5000 µg/mL. EAE-Ph was the most effective extract, being active against all tested strains, except *Pseudomonas aeruginosa* (ATCC 27853). EE-Ph was less effective, being not active against three of the 14 strains tested. HE-Ph was not active against all the strains in the concentration gradient tested.

Regarding the antibacterial effect, EAE-Ph MIC value of 5000 µg/mL for *P. aeruginosa* (ATCC 9027) was bactericidal, while MIC values of EAE-Ph and EE-Ph for all other strains tested were classified as bacteriostatic.

The antibiotics (AMP and CHL) MIC values are in accordance with CLSI document, validating the assay. AMP and CHL were selected in accordance with the CLSI document that describes the antimicrobials that should be reported to the bacterial species tested. Furthermore, AMP was selected to represent a simple penicillin being able to be a marker to resistance mechanisms, such as the production of beta-lactamases. CHL was selected to represent a non-conventional therapeutic option to the treatment of bacterial infections that possess some resistance to other antimicrobials, but still sensible to this one. In addition, the different mechanisms of action of AMP (action in the bacterial cellular membrane) and CHL (action in the protein synthesis) were considered in the selection process.

As previously mentioned, there are very few reports on the antibacterial activity of *P. hydropiperoides*, but the existing ones corroborate our findings. Bussmann et al. [37] when analyzing ethanolic and aqueous extracts of this plant species against *S. aureus* (ATCC 25923) by the broth microdilution method, reported activity of the ethanolic extract against this strain, with a MIC value of 1000 µg/mL, considering it a promising extract. Miranda et al. [38] also evaluated the antibacterial activity of the ethanolic and aqueous extracts of *P. hydropiperoides* against *S. aureus* (ATCC 33591) by the diffusion method according to Kirby and Bauer and reported that the ethanolic extract was active against this strain in all alcoholic strengths tested, but mainly in 60% and 100%, at 200 and 500 mg/mL. Bouzada et al. [39] reported the activity of the methanol extract of *P. hydropiperoides* by the broth microdilution method with MIC values equal to 5000 µg/mL (leaves) and 2500 µg/mL (flowers) for *S. aureus* (ATCC 6538) and 2500 µg/mL (leaves and flowers) for *Salmonella* Typhimurium (ATCC 13311).

*S. aureus* is a Gram-positive coccus species related to community and healthcare-associated infections (HAIs), being responsible for skin and soft tissues infections, food poisoning, and the most complicated clinical situations, such as endocarditis, osteomyelitis, pneumonia, bloodstream infections, and septic shock syndrome [40]. The most complex infections caused by *S. aureus* are related to the resistance genes expression, especially the *mecA*, that concedes the mutation on the penicillin-binding protein 2a (PBP2a), promoting the lower affinity to β-lactams such as penicillin, amoxicillin, and methicillin [41]. Considering the MRSA strains, Dzoyem et al. [42] described the antibacterial activity of *Polygonum limbatum* against MRSA (ATCC 33591). Zuo et al. [43] previously reported the anti-MRSA activity of *Polygonum multiflorum* (MIC ≤ 1.43 mg/mL) against nine clinical MRSA isolates and methicillin-sensitive *S. aureus* (ATCC 25923) (MSSA), corroborating our findings. In the present study, five MRSA isolated from blood cultures were tested and MIC values ranged from 625 to 1250 µg/mL to EAE-Ph and were equal to 1250 µg/mL to EE-Ph. According to Simões, Bennett, and Rosa [44], phytochemicals are routinely classified as antimicrobials based on susceptibility tests that produce MIC values of 100 to 1000 µg/mL. On the other hand, Kuete’s [45] classification to plant extracts antimicrobial activity demonstrates a moderate antibacterial effect (100 < MIC ≤ 625 µg/mL) of EAE-Ph phytochemicals against MRSA 1664534 (MIC equal to 625 µg/mL) and a weak activity (MIC > 625 µg/mL) of EAE-Ph and EE-Ph against the other strains tested. Probably, this activity is related to galloyl-glucose derivatives as previously discussed, based on Pawłowska et al. [29] reports.

In addition, it has been shown that the combination of antibiotics and extracts can be more effective than the isolated substances [40] and the synergism between antibiotics (such as β-lactams and aminoglycosides or fluoroquinolones) has been used in clinical practice [46]. This is an exciting field of research, enabling available antibiotics, no longer effective due to developed bacterial resistance, to be useful again while in combination with phytochemicals, which themselves cannot be used in monotherapy due to their higher MIC (100–5000 µg/mL) [24].

## 3. Materials and Methods

### 3.1. Chemicals and Reagents

All chemicals and reagents used, including solvents, were analytical or HPLC grade as follows: hexane P.A., ethyl acetate P.A. and ethanol P.A. (Neon^®^ Comercial, São Paulo, Brazil); Müeller-Hinton agar (MHA) and Müeller-Hinton broth (MHB) (Difco^®^ Laboratories, Detroit, MI, USA); ampicillin, chloramphenicol, iodonitrotetrazolium chloride, quercetin, rutin, gallic acid, and BHT (Sigma-Aldrich^®^, St. Louis, MO, USA). Purified water was obtained using the reverse osmosis water system (Permution^®^, Paraná, Brazil).

### 3.2. Plant Material

Aerial parts (stems, leaves, and inflorescences) of *P. hydropiperoides* were collected at the Medicinal Garden of the Faculty of Pharmacy, Federal University of Juiz de Fora, Juiz de Fora, Minas Gerais, southeast region of Brazil (21°46′40″ South, 43°22′1″ West) on 22 November 2016. The species was identified by Dr. Fátima Regina Gonçalves Salimena and a voucher specimen (CESJ no. 46.072) was deposited in the Herbarium Leopoldo Krieger, Federal University of Juiz de Fora. The plant name *Polygonum hydropiperoides* Michx. has been checked with http://www.worldfloraonline.org (accessed on 18 February 2023), being synonym of *Persicaria hydropiperoides* (Michx.) Small [47]. This study is duly registered in the Brazilian National System for the Management of Genetic Heritage and Associated Traditional Knowledge (SisGen) under registration number ACEAD8C.

### 3.3. Extraction Procedures

Selected aerial parts (458 g) were dried in an oven, at 40 °C, with forced air until there was a humidity loss of 90 to 95% [48]. After this, the vegetable material was cut with an industrial knife mill, pulverized on tamis (n. 20), and subjected to static maceration extraction at room temperature with solvents of increasing polarity (hexane, ethyl acetate, and ethanol).

Initially, the pulverized plant material was extracted with hexane P.A. (hexane extract, HE-Ph). The resulting solution was filtered on simple filter paper and concentrated on a rotary evaporator (R-215 Büchi Labortechnik AG, Flawil, Switzerland) at 40 °C. The same procedure was performed over the solid residue with ethyl acetate P.A. (ethyl acetate extract, EAE-Ph) and, subsequently, with ethanol P.A. (ethanolic extract, EE-Ph). From 458 g of dried and pulverized plant material subjected to extraction, it is observed that EE-Ph showed the highest yield (13.9%; 63.5 g), followed by EAE-Ph (1.3%; 5.8 g) and HE-Ph (0.6%; 2.8 g). The three extracts were stored under refrigeration, between 2 and 8 °C and protected from light, until the time of use in chemical and biological tests. The hexane extract was only used in this work in the microbiological assays.

### 3.4. Chromatographic Analysis

The phytochemical profile was evaluated using high-performance liquid chromatography with a diode array detector and an ion trap mass spectrometer with an electrospray interface (HPLC-DAD-ESI/MS^n^). About 5–10 mg of EAE-Ph and EE-Ph were re-dissolved in 1 mL of methanol, filtered through 0.45 μm nylon filters, and 10 μL of sample was injected. The HPLC system was an Agilent Series 1100 with a G1315B diode array detector and an ion trap mass spectrometer (Esquire 6000, Bruker^®^ Daltonics, Billerica, MA, USA) with an electrospray interface. Compounds were separated with a Phenomenex Gemini C_18_ column (5 μm, 250 × 3.0 mm i.d.; Phenomenex, CA, USA), at 30 °C. Chromatographic conditions included a solvent gradient system [acetonitrile (A) and water/formic acid (100/0.1, *v*/*v*) (B): 20% A (0 min), 25% A (10 min), 25% A (20 min), 50% A (40 min), 100% A (42–47 min), 20% A (49–55 min)], with a flow adjusted to 0.4 mL/min, as described by Gouveia-Figueira and Castilho [49]. DAD chromatograms were recorded at wavelengths of 280, 320, and 350 nm. The most abundant compounds were quantified using the following analytical standards: quercetin (350 nm) for flavonoids (except catechin and its derivatives, which were quantified using catechin at 280 nm), and gallic acid for phenolic acids at 320 nm. Calibration graphs were constructed in the 0.5–100 mg/L range. Peak areas were plotted vs analyte concentration. Detection limits (3σ criterion) were 0.1–0.2 mg/L. Repeatability (*n* = 10) and intermediate precision (*n* = 9, three consecutive days) were lower than 4 and 8%, respectively. Chromatograms at each wavelength for the corresponding extracts are shown in Appendix A.

### 3.5. In Vitro Antioxidant Activity

#### 3.5.1. Phosphomolybdenum Reducing Power

The total antioxidant capacity of EAE-Ph and EE-Ph was evaluated by phosphomolybdenum reducing power assay according to Prieto et al. [50], using ascorbic acid as the standard. Gallic acid and quercetin were tested as reference antioxidant substances. This spectrophotometric method is based on the reduction in molybdenum (VI) to molybdenum (V) in the presence of certain substances with antioxidant capacity, with formation of phosphate/molybdenum (V) green complex at acid pH [49]. The extract solutions (300 µL, or 97.82 µg/mL) were added to 2000 μL of the phosphomolybdenic complex reagent and kept in a water bath at 95° C for 90 min. After cooling, 250 μL was transferred into a 96-well microplate, and absorbance was measured in a microplate reader (Thermo Scientific^™^, Waltham, MA, USA) at 695 nm. The experiment was performed in triplicate. The results were expressed as ascorbic acid equivalent (AAE) ± standard deviation (SD).

#### 3.5.2. Nitric Oxide (NO) Inhibition

The antioxidant activity of EAE-Ph and EE-Ph was also evaluated by the indirect production of NO by Griess’ method, according to Green et al. [51], with few adjustments. For NO production, sodium nitroprusside (NPS) (10 mM) in a phosphate buffer (PBS) (10 mM, pH 7.4) was used. Samples (EAE-Ph and EE-Ph) and standards (gallic acid and Trolox—vitamin E analogue) solutions in PBS were prepared, with 1% dimethyl sulfoxide (DMSO), at concentrations ranging from 25 to 400 μg/mL. The assay was performed in 96-well microplates that were incubated in the presence of light at room temperature for 60 min, followed by the addition of the Griess reagent (1% sulfanilamide + 0.1% N-(1-naphthyl)-ethylenediamine (NED) in 2.5% phosphoric acid). After 10 min, the absorbance was measured at a wavelength of 540 nm in a microplate reader (Thermo Scientific^™^, MA, USA). Appropriate controls were performed. The experiment was carried out in triplicate. Results were expressed in percentage of inhibition of NO production ± standard deviation (SD).

#### 3.5.3. β-Carotene Bleaching Assay

The antioxidant activity of EAE-Ph and EE-Ph was additionally investigated by β-carotene/linoleic acid system method as described by Koleva et al. [52], with little adjustments. EAE-Ph and EE-Ph (250 µg/mL) and quercetin, rutin, and BHT (25 µg/mL) ethanolic solutions were prepared. The β-carotene/linoleic acid emulsion was made in a round bottom flask protected from light with aluminum foil where were added 1000 μL of β-carotene in chloroform solution (0.20 mg/mL), linoleic acid (25 μL), and Tween 40 (200 mg). After mixing, the chloroform was evaporated using a rotary evaporator (R-215 Büchi Labortechnik AG, Flawil, Switzerland) at 40° C, and distilled water previously saturated with oxygen for 30 min was added (50,000 μL). Then, 30 μL of EAE-Ph, EE-Ph, quercetin, rutin, and BHT solutions were placed into a 96-well microplate. Then, 250 μL of the emulsion were added and the microplate was maintained at 50 °C for 105 min. The reaction was monitored by discoloration of β-carotene by absorbance reduction measurement at 492 nm, with reading at 15 min intervals for a total of 105 min, using a microplate reader (Thermo Scientific^™^, MA, USA). Appropriated controls were performed. The experiment was made in quadruplicate. The results were expressed in percentage of inhibition of lipid peroxidation (%I) ± standard deviation (SD).

### 3.6. In Vitro Antibacterial Activity

The antibacterial activity was determined by the minimal inhibitory concentration (MIC) using the broth microdilution method according to Clinical and Laboratory Standards Institute (CLSI) M07-A10 [53] and M100-S30 [36] documents. The determination of minimal bactericidal concentration (MBC), followed by the classification of the antibacterial effect as bactericidal or bacteriostatic procedures, was carried out according to Andrews [54]. Nine American Type Culture Collection (ATCC^®^) bacterial strains of methicillin-sensitive *Staphylococcus aureus* subsp. *aureus* Rosenbach (ATCC 6538, ATCC 25923 and ATCC 29213) (MSSA), *Escherichia coli* (Migula) Castellani and Chalmers (ATCC 10536 and ATCC 25922), *Salmonella enterica* subsp. *enterica* (*ex* Kauffmann and Edwards) Le Minor and Popoff serovar Choleraesuis (ATCC 10708), *Salmonella enterica* subsp. *enterica* (*ex* Kauffmann and Edwards) Le Minor and Popoff serovar Typhimurium (ATCC 13331), and *Pseudomonas aeruginosa* (Schroeter) Migula (ATCC 9027 and ATCC 27853), described as requested by ATCC^®^, were assayed.

In addition, five clinical methicillin-resistant *S. aureus* strains (MRSA 1485279, MRSA 1605677, MRSA 1664534, MRSA 1688441, and MRSA 1830466) isolated from a blood culture and identified with VITEK2^®^ in the Clementino Fraga Filho Universitary Hospital of the Federal University of Rio de Janeiro and kindly provided by MSc. Adriana Lúcia Pires Ferreira, were tested.

#### Minimal Inhibitory Concentration (MIC) and Minimal Bactericidal Concentration (MBC) Determinations

As previously described, to determine the MIC values the broth microdilution method was applied according to CLSI M07-A10 guideline [53]. HE-Ph, EAE-Ph, and EE-Ph stock solutions were prepared using water and DMSO, respecting the limit of 1% of this organic solvent in the first well of the microtitration plate [36], at the concentration of 10 mg/mL. To validate the assay, ampicillin (AMP) and chloramphenicol (CHL) were used as control antibiotics. The antibiotic solutions were prepared at a concentration of 1 mg/mL with their respective solvents/diluents, as described in document M100-S30 [36].

In a sterile 96-well microplate, twofold serial dilutions of extracts (quadruplicate) and antibiotics (triplicate) were prepared in MHB at concentrations ranging from 5000 to 40 μg/mL and 500 to 4 μg/mL, respectively. MIC values above 5000 μg/mL were not determined. Subsequently, 10 μL of standardized bacteria suspension according to the 0.5 McFarland scale were added. After incubation at 35 ± 2 °C for 16 to 20 h under aerobic conditions, 20 μL of 1 mg/mL iodonitrotetrazolium chloride (INT) solution (*w*/*v*) was used as an indicator of bacterial growth (any color change from purple to pink was recorded as bacterial growth). Then, this system was incubated for a further 30 min at the same conditions, and the MIC was determined. The appropriate controls were performed. After the determination of MIC values, MBC was established according to Andrews’ method [54] by the spreading of 10 μL of suspensions from wells showing no visible bacterial growth on MHA Petri dishes. After incubation at 35 ± 2 °C for 16 to 20 h under aerobic conditions, the presence or absence of bacterial growth was analyzed. MBC was determined as the lowest concentration of dilutions that prevented the visible bacterial growth after subculture on MHA Petri dishes. Bacterial growth or no bacterial growth on MHA revealed a bacteriostatic or bactericidal effect, in this order.

### 3.7. Statistical Analysis

The statistical analysis was performed using GraphPad Prism 7.0 software, data were submitted to analysis of variance (ANOVA) followed by Tukey’s test to measure the degree of significance for *p* < 0.05. Results were expressed as mean ± standard deviation (SD).

## 4. Conclusions

The results reported in the present study suggest that *Polygonum hydropiperoides* is a natural source of active substances, such as phenolic acids and flavonoids, with particular relevance for galloyl-containing compounds, especially in the ethyl acetate extract (EAE-Ph). Gallic acid and β-glucogallin can act in synergic mode since the extract shows higher antioxidant capacity than gallic acid alone. These results add scientific support for the popular use of *Polygonum hydropiperoides* in the treatment of inflammatory processes and wounds.

## Figures and Tables

**Figure 1 plants-12-01606-f001:**
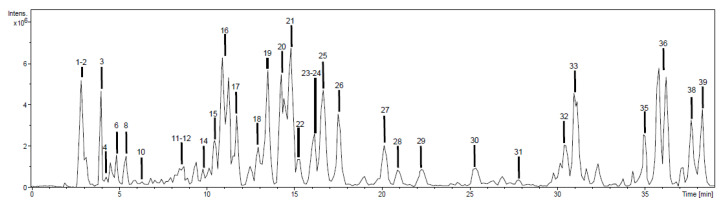
Chromatographic profiling of ethyl acetate extract from aerial parts of *Polygonum hydropiperoides* (EAE-Ph).

**Figure 2 plants-12-01606-f002:**
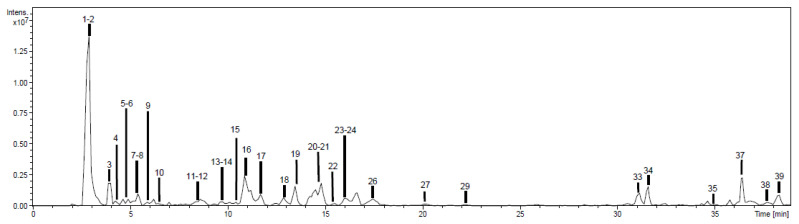
Chromatographic profiling of ethanolic extract from aerial parts of *Polygonum hydropiperoides* (EE-Ph).

**Table 1 plants-12-01606-t001:** Concentration (mg/g of dry extract) of main phenolic compounds in ethyl acetate (EAE-Ph) and ethanolic (EE-Ph) extracts from aerial parts of *Polygonum hydropiperoides*.

No.	Assigned Identification	EAE-Ph	EE-Ph
*Phenolic Acids*			
2	Galloyl glucose	43.00 ± 3.00	48.00 ± 1.00
3	Gallic acid	21.00 ± 1.00	4.20 ± 0.01
29	Gallic acid derivative	0.26 ± 0.01	--
Total		64.26 ± 4.01	52.20 ± 1.01
*Flavonoids*			
6	Procyanidin dimer	3.40 ± 0.10	--
8	Catechin	4.10 ± 0.20	1.10 ± 0.10
15	Myricetin-*O*-deoxyhexoside	0.15 ± 0.02	--
16	Quercetin-*O*-hexoside	4.00 ± 0.20	0.68 ± 0.01
17	(Epi)catechin-*O*-gallate	4.80 ± 0.50	1.07 ± 0.08
18	Isorhamnetin-*O*-rutinoside	0.38 ± 0.03	0.08 ± 0.01
19	Quercetin-*O*-pentoside	1.60 ± 0.10	0.19 ± 0.01
20 + 21	Quercetin glycosides	6.30 ± 0.30	0.73 ± 0.01
22	Quercetin-*O*-acetylhexoside	0.32 ± 0.03	0.01 ± 0.00
27	Isorhamnetin-*O*-deoxyhexoside	0.45 ± 0.01	0.01 ± 0.00
28	Quercetin-*O*-acetylpentoside	0.25 ± 0.02	--
30	Quercetin-*O*-acetylpentoside	0.31 ± 0.03	--
32	Mearnsetin	0.44 ± 0.05	--
33	Quercetin	2.35 ± 0.03	0.18 ± 0.01
39	Isorhamnetin	2.11 ± 0.05	0.17 ± 0.01
Total		30.96 ± 1.67	4.22 ± 0.24
TIPC		95.22 ± 5.68	56.42 ± 1.25

TIPC: total individual phenolic content (sum of all the quantified individual phenolics); data represent mean ± standard deviation (*n* = 3).

**Table 2 plants-12-01606-t002:** In vitro antioxidant activity of the ethyl acetate (EAE-Ph) and ethanolic (EE-Ph) extracts from aerial parts of *Polygonum hydropiperoides* and standards by phosphomolybdenum reducing power, nitric oxide inhibition, and β-carotene bleaching assays.

Sample	Phosphomolybdenum Reducing Power	Inhibition of Nitric Oxide Production (%)					β-Carotene
	AAE (µg/mg)	400 µg/mL	200 µg/mL	100 µg/mL	50 µg/mL	25 µg/mL	Inhibition of Lipid Peroxidation (%)
EAE-Ph	214.57 ± 23.99 ^a,^*^,#^	67.96 ± 3.22 ^†,‡^	66.30 ± 3.33 ^†,‡^	56.86 ± 3.11 ^†,‡^	45.06 ± 5.58 ^e^	37.48 ± 9.13 ^g^	74.44 ± 5.17 ^¤,Φ,i^
EE-Ph	230.82 ± 4.60 ^a,^*^,#^	47.56 ± 1.71 ^b^	43.05 ± 1.60 ^c^	33.73 ± 2.35 ^d^	22.36 ± 1.49 ^f^	15.31 ± 2.95 ^h^	45.56 ± 8.62 ^Φ,¥,j^
Gallic acid	160.78 ± 17.04	37.62 ± 7.25 ^b^	33.96 ± 2.43 ^c^	31.94 ± 1.33 ^d^	29.70 ± 0.11 ^e,f^	25.99 ± 1.87 ^g,h^	-
Quercetin	424.70 ± 10.99	-	-	-	-	-	43.33 ± 5.47 ^j^
Trolox	-	42.43 ± 8.69 ^b^	37.25 ± 9.25 ^c^	32.79 ± 7.68 ^d^	29.93 ± 10.69 ^e,f^	26.92 ± 8.73 ^g,h^	-
Rutin	-	-	-	-	-	-	27.78 ± 7.43
BHT	-	-	-	-	-	-	61.11 ± 5.33 ^i^

AAE: ascorbic acid equivalent. Data represent mean ± standard deviation (*n* = 3, for phosphomolybdenum reducing power and inhibition of NO; *n* = 4, for β-carotene bleaching assay). Statistical analysis performed by ANOVA test followed by Tukey’s test. In phosphomolybdenum reducing power: consider significantly different values from gallic acid for ** p* < 0.01, and significantly different values from quercetin for *# p* < 0.01; In inhibition of NO production: for each concentration, consider significantly different values of gallic acid for ^†^
*p* < 0.01 and significantly different values from trolox for ^‡^
*p* < 0.01; In β-carotene bleaching assay: significantly different values from rutin for ^¤^
*p* < 0.01, significantly different values from quercetin for ^Φ^
*p* < 0.01 and significantly different values from BHT for ^¥^
*p* < 0.01. In all experiments, equal letters represent significantly equal values between samples (*p* > 0.05).

**Table 3 plants-12-01606-t003:** Minimal inhibitory concentration (MIC) and minimal bactericidal concentration (MBC) values obtained for the hexane (HE-Ph), ethyl acetate (EAE-Ph), and ethanolic (EE-Ph) extracts from the aerial parts of *Polygonum hydropiperoides*, and ampicillin (AMP) and chloramphenicol (CHL) against the bacterial strains tested.

Bacterial Strain	MIC (µg/mL)					MBC (µg/mL)		
	HE-Ph	EAE-Ph	EE-Ph	AMP	CHL	HE-Ph	EAE-Ph	EE-Ph
*Staphylococcus aureus* (ATCC 6538)	>5000	1250 ^2^	1250 ^2^	<4 ^a^	8 ^c^	>5000	5000	5000
*Staphylococcus aureus* (ATCC 29213)	>5000	1250 ^2^	1250 ^2^	<4 ^a^	16 ^c^	>5000	5000	5000
*Staphylococcus aureus* (ATCC 25923)	>5000	1250 ^2^	1250 ^2^	<4 ^a^	16 ^c^	>5000	>5000	5000
*Escherichia coli* (ATCC 10536)	>5000	5000 ^2^	>5000	<4 ^b^	4 ^c^	>5000	>5000	>5000
*Escherichia coli* (ATCC 25922)	>5000	5000 ^2^	>5000	<4 ^b^	4 ^c^	>5000	>5000	>5000
*Salmonella* Choleraesuis (ATCC 10708)	>5000	5000 ^2^	5000 ^2^	<4 ^b^	4 ^c^	>5000	>5000	>5000
*Salmonella* Typhimurium (ATCC 13311)	>5000	5000 ^2^	5000 ^2^	<4 ^b^	4 ^c^	>5000	>5000	>5000
*Pseudomonas aeruginosa* (ATCC 9027)	>5000	5000 ^1^	5000 ^2^	>500	8	>5000	5000	>5000
*Pseudomonas aeruginosa* (ATCC 27853)	>5000	>5000	>5000	>500	8	>5000	>5000	>5000
MRSA 1485279	>5000	1250 ^2^	1250 ^2^	>500 ^a^	125 ^c^	>5000	2500	2500
MRSA 1605677	>5000	1250 ^2^	1250 ^2^	>500 ^a^	8 ^c^	>5000	5000	5000
MRSA 1664534	>5000	625 ^2^	1250 ^2^	32 ^a^	16 ^c^	>5000	2500	5000
MRSA 1688441	>5000	1250 ^2^	1250 ^2^	500 ^a^	8 ^c^	>5000	5000	2500
MRSA 1830466	>5000	1250 ^2^	1250 ^2^	250 ^a^	16 ^c^	>5000	2500	5000

MRSA: methicillin-resistant *Staphylococcus aureus*. According to CLSI document M100-S30 [36] the MIC values for: AMP (Penicillin) (a) ≤0.12 and ≥0.25 µg/mL classify this bacteria as sensitive and resistant, respectively; AMP (b) ≤8, 16 e ≥ 32 µg/mL classify this bacteria as sensitive, intermediate, and resistant, in this order; CHL (c) ≤8, 16 e ≥ 32 µg/mL classify this bacteria as sensitive, intermediate, and resistant, respectively; 1—bactericidal effect; 2—bacteriostatic effect.

## Data Availability

Not applicable.

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
