# Peer review of "Phenolics Profiling by HPLC-DAD-ESI/MSn of the Scientific Unknown Polygonum hydropiperoides Michx. and Its Antioxidant and Anti-Methicillin-Resistant Staphylococcus aureus Activities"

_plants, 2023, doi:10.3390/plants12081606_

Round 1

Reviewer 1 Report (Previous Reviewer 3)

Dear authors,  After reviewing the following manuscript entitled "Phenolics profiling by HPLC-DAD-ESI/MSn of the scientific unknown Polygonum hydropiperoides Michx and its antioxidant and anti-methicillin-resistant Staphylococcus aureus activities" and with reference number (Plants - 2276276), I sent the following comments and observations that the authors should attend to before its publication in this journal. I appreciate the work of the authors, but please resolve the following data:

In the introduction, it would be good to specify more precisely the results obtained when determining the biological activities pursued in the study.

In Chemical characterization it is not relevant how TIPC was performed. Methods for the determination of total polyphenols (E.g. Folin-Ciocâlteu method) and total flavonoids (the colorimetric method) should be applied.  Knowing that the solvents used in the extraction also have antimicrobial properties, I recommend that it be mentioned how they influenced the antimicrobial activity of the studied extracts.  It should also be specified on which criteria ampicillin and chloramphenicol were chosen in the study  In table 3, it is not clearly specified in what concentration the antibiotics were used.

The old references must be removed.

Author Response

We are grateful to the reviewers for their suggestions and comments that will improve our manuscript. We carefully read all the contributions and analyzed one by one to make the suitable adjustments. Below you will find the reviewers comments and our answers to them.

REVIEWER 1

Comment 1: “In the introduction, it would be good to specify more precisely the results obtained when determining the biological activities pursued in the study”.

Answer 1: We appreciate the comment and improved the Introduction section with a new paragraph providing background about all the studied biological activities. On the other hand, we kindly disagree to write about results of the study in the Introduction. We think that is more appropriate to present and discuss about the results only in the Results and Discussion section, because the Plants journal instruction for author describes the Introduction section as: “The introduction should briefly place the study in a broad context and highlight why it is important. It should define the purpose of the work and its significance, including specific hypotheses being tested. The current state of the research field should be reviewed carefully and key publications cited. Please highlight controversial and diverging hypotheses when necessary. Finally, briefly mention the main aim of the work and highlight the main conclusions. Keep the introduction comprehensible to scientists working outside the topic of the paper”.

Comment 2: “In Chemical characterization it is not relevant how TIPC was performed. Methods for the determination of total polyphenols (E.g. Folin-Ciocâlteu method) and total flavonoids (the colorimetric method) should be applied”.

Answer 2: We kindly disagree about this comment. Colorimetric methods have a large range of interferences being able to overestimate the content of phenolic acids and flavonoids (for instance TPC, being based on a redox reaction, gives positive results in the presence of reducing sugars). So, we performed a quantification of these metabolites using High-Performance Liquid Chromatography (HPLC) because it is an analytical tool very precise and reliable.

Comment 3: “Knowing that the solvents used in the extraction also have antimicrobial properties, I recommend that it be mentioned how they influenced the antimicrobial activity of the studied extracts”.

Answer 3: As described in the 3.6.1 section (line 523), the appropriate controls were performed to guarantee the obtained results. Among them, the solvent/diluent controls were performed and showed that they did not interfere in the bacterial growth. So, the found activity is related only to the metabolites present in the tested extracts. However, a new paragraph about this was written in the Results and Discussion section (lines 314-317). We think this should solve the problem.

Comment 4: “It should also be specified on which criteria ampicillin and chloramphenicol were chosen in the study”.

Answer 4: We appreciate this suggestion and a new paragraph discussing the selection criteria was added in the manuscript (lines 327-334).

Comment 5: “In table 3, it is not clearly specified in what concentration the antibiotics were used”.

Answer 5: As mentioned in the section 3.6.1, twofold serial dilutions were performed creating a concentration gradient of the tested samples [extracts (5,000 to 40 µg/mL) and antibiotics (500 to 4 µg/mL)] as described in Material and Methods. So, the results present in Table 3 are the minimal inhibitory concentration (MIC) values obtained in the assays against the bacterial strains tested.

Comment 6: “The old references must be removed”.

Answer 6: We appreciate the care about the references, but the old ones are related to the validated methods used in the present study (primary sources) and/or about the plant species, since Polygonum hydropiperoides is a scarcely scientific exploited species.

Kind regards,

The manuscript authors.

Reviewer 2 Report (Previous Reviewer 4)

The article describes the result of the chemical characterization of ethyl acetate, ethanolic and hexane extract from aerial parts of Polygonum hydropiperoides. Mentioned extracts were analyzed to determine the profile of phytochemicals and antibacterial and antioxidant activity.

Paragraph 3.4 please add some information about the quantitative analysis if it was performed.

Paragraph 3.5 , 3.6 please add information how many repetitions was taken in such analysis.

Why the statistical analysis is not described in the section materials and methods?

Conclusion: Please do not use the abbreviation in this paragraph.

Author Response

We are grateful to the reviewers for their suggestions and comments that will improve our manuscript. We carefully read all the contributions and analyzed one by one to make the suitable adjustments. Below you will find the reviewers comments and our answers to them.

REVIEWER 2

Comment 1: “Paragraph 3.4 please add some information about the quantitative analysis if it was performed”.

Answer 1: We are grateful for the care about this section and we added more specific information about it (lines 431-437).

Comment 2: “Paragraph 3.5 , 3.6 please add information how many repetitions was taken in such analysis”.

Answer 2: The number of repetitions in all the assays are described in Material and Methods sections. However, triplicates or quadruplicates in independent experiments were performed in the assays (lines 450, 466-467, 483-484, 515-516).

Comment 3: “Why the statistical analysis is not described in the section materials and methods?”.

Answer 3: We are sorry for forgetting this section and we are most grateful for the reviewer signalizing the omission. So, a new section 3.7 (lines 532-535) was added.

Comment 4: “Conclusion: Please do not use the abbreviation in this paragraph”.

Answer 4: We appreciate the suggestion and rewrote the conclusion without the abbreviations.

Kind regards,

The manuscript authors.

Round 2

Reviewer 1 Report (Previous Reviewer 3)

I believe that the observations have been fixed and I agree with the publication.

Author Response

Dear Reviewer 1,

Thank you very much for your precious contributions to improve our manuscript.

Kind regards,

The manuscript authors.

This manuscript is a resubmission of an earlier submission. The following is a list of the peer review reports and author responses from that submission.

Round 1

Reviewer 1 Report

Figures and tables should be submitted as supplementary material.

Reviewer 2 Report

Costa and co-workers present the manuscript entitled “Phenolics profiling by HPLC-DAD-ESI/MSn and the in vitro antibacterial activity of the scientific unknown Polygonum hydropiperoides Michx”,  where they chemically characterize and investigate the antibacterial activity of extracts of different polarities from aerial parts of P. hydropiperoides.

The authors present a very good metabolomic analysis of extracts of different polarities obtained from P. hydropiperoides, which adds value to the field. However, as found by the authors in the antibacterial screening, and also corroborated by previous reports, the activity P. hydropiperoides is to disregard, considering the excessively high MIC values found. In this sense, the present survey gets undervalued, being a report of the chemical composition alone.

Once I consider it has potential for publication, I would suggest the authors to explore the antioxidant potential of the extracts in a physiologically relevant free radical, for instance superoxide anion radical, and to perform a statistical correlation between the chemical composition and the bioactivity. This will definitely valorize the study and the interest in P. hydropiperoides exploitation.

Other issues should be taken into account:

Metodology

  1. The authors used the aerial parts of P. hydropiperoides in this survey. In order to infer about the species representability, the authors must mention the number of individuals used for the preparation of the extracts. Please add this information to the methodology section.
  2. The “selected aerial parts” of the plant used in the preparation of the extracts should be specified. Did the authors extract stems, flowers and leaves? Also, the particle size after pulverization should be mentioned. Please add this information to the manuscript.
  3. The calibration curves of the authentic standards used for compounds identification and quantification must be displayed.

Results

  1. Upon compounds characterization based on their mass fragmentation patterns, the authors frequently state that “compound XX was characterized as YY”. Once the authors did not analyse authentic standards for all the compounds, they should review the whole manuscript and rewrite “tentatively identified as” instead of “characterized as”, in this situations. Moreover, a reference comprising mass fragmentation must be added every time a tentatively identification is performed with no available standard.
  2. Lines 178-180: please complete the sentence
  3. Line 206: “to the ability against ATCC® bacterial strains”; please rephrase

Reviewer 3 Report

I appreciate the work of the authors on the physico-chemical characterization by HPLC method.

It would have been interesting if more extraction methods had been applied using plant/solvent ratio variants.Thus, a more sustainable analysis of the chemical composition and extraction of bioactive compounds could be performed.To evaluate the antimicrobial activity, it would have been good if antibiotic molecules from the new generations were studied.Therefore, I propose to deepen the study and thus encourage its resubmission.

Reviewer 4 Report

I think that the paper is interesting and presents new information about the bioactive compunds that can be found in the P. hydropiperoides and also its antibacterial activity.